# Regulation of KIR3DL3 Expression via miRNA

**DOI:** 10.3390/genes10080603

**Published:** 2019-08-09

**Authors:** Rungtiwa Nutalai, Silvana Gaudieri, Amonrat Jumnainsong, Chanvit Leelayuwat

**Affiliations:** 1Biomedical Sciences Program, Graduates School of Khon Kaen University, Khon Kaen 40002, Thailand; 2Department of Clinical Immunology and Transfusion Sciences, School of Medical Technology, Faculty of Associated Medical Sciences, Khon Kaen University, Khon Kaen 40002, Thailand; 3The Centre for Research and Development of Medical Diagnostic Laboratories (CMDL), Faculty of Associated Medical Sciences, Khon Kaen University, Khon Kaen 40002, Thailand; 4School of Human Sciences, University of Western Australia, Perth, WA 6009, Australia; 5Institute for Immunology and Infectious Diseases, Murdoch University, Perth, WA 6150, Australia; 6Division of Infectious Diseases, Vanderbilt University Medical Center, Nashville, TN 37232, USA

**Keywords:** *killer-cell immunoglobulin-like receptor (KIR) 3DL3*, miRNA, 3’untranslated region, post-transcriptional regulation

## Abstract

Killer-cell immunoglobulin-like receptor (KIR) 3DL3 is a framework gene present in all human KIR haplotypes. Although the structure of KIR3DL3 is suggestive of an inhibitory receptor, the function of KIR3DL3 has not been demonstrated and cognate ligands have not been identified. KIR3DL3 has been shown to be constitutively expressed at a low RNA level in peripheral blood mononuclear cell (PBMC) and decidual natural kill (NK) cells, but cell surface expression of KIR3DL3 cannot be detected. Accordingly, post-transcriptional regulation of KIR3DL3 should exist. Using bioinformatics analysis, we identified three candidate micro ribonucleic acids (miRNAs; miR-26a-5p, -26b-5p and -185-5p) that potentially regulate KIR3DL3 expression. Luciferase reporter assays utilizing constructs with mutated miRNA-binding sites of miR-26a-5p, -26b-5p and -185-5p in the 3’-untranslated region (3’ UTR) of KIR3DL3 resulted in up-regulation of luciferase activity demonstrating a potential mechanism of gene regulation. Furthermore, knockdown of the same endogenous miRNAs using silencing ribonucleic acid (siRNA) led to induced surface expression of KIR3DL3. In conclusion, we provide a novel mechanism of functional regulation of KIR3DL3 via miRNAs. These findings are relevant in understanding the generation of KIR repertoire and NK cell clonality.

## 1. Introduction

Killer-cell immunoglobulin-like receptors (KIRs) are transmembrane glycoproteins consisting of both inhibitory and activating receptors on natural kill (NK) cells and on a subset of T cells [1,2]. NK cells exhibit at least one inhibitory receptor to discriminate cells with aberrant human leucocyte antigen (HLA) class I surface expression often observed during infection and malignant transformation from normal cells [3]. This receptor-ligand interaction is important in establishing the NK cell activation threshold and is influenced by the diversity in both sets of molecules. KIR diversity is generated through: (I) Allelic variations, (II) the level of surface expression, and (III) haplotypic diversity by the presence or absence of KIR genes. Furthermore, each individual NK cell can express different KIR repertoires ranging from one to eight receptors to generate NK clonality [4,5]. For example, four conserved KIR genes are present in virtually all individuals, known as the framework genes KIR3DL3, KIR3DP1, KIR2DL4, and KIR3DL2 [6,7], but only KIR2DL4 is expressed at both the RNA and protein level in all individuals [4,8]. It is still unclear how different KIR repertoires are generated from the same genetic template but the distribution of KIRs appears to be partly determined by DNA methylation releasing a broad range of functional responses [9,10,11]. 

KIR3DL3, also known as KIRC1, KIR44, and KIR3DL7, is a member of the inhibitory KIR surface receptors possessing three extracellular immunoglobin (Ig) domains (3D) and only one inhibitory motif within the long cytoplasmic domain (L) due to a premature stop codon [12,13,14]. Moreover, exon 6 of KIR3DL3 encoding the stem part of the receptor is also absent, which differs from other inhibitory KIRs. Understanding the function of KIR3DL3 is further complicated by the unknown identity of the specific ligand, but its importance is highlighted by its presence in all human KIR haplotypes with 120 distinct polymorphic alleles of the coding sequence [12,15]. 

An unusual feature of KIR3DL3 is its limited detection at the mRNA level as reflected by its methylation status [12,16,17], even though the KIR3DL3 promoter shows the strongest activity of all KIR genes [17]. However, KIR3DL3 mRNA expression has been recently identified in decidual NK cells and in the cluster of differentiation (CD) 56^bright^ NK cell subset in peripheral blood mononuclear cells (PBMCs) with females showing higher levels of expression than males, but there is a lack of KIR3DL3 protein expression [16]. Accordingly, it has been proposed that KIR3DL3 is expressed in specific pathological situations or during development. However, regulation of KIR3DL3 mRNA at the post-transcriptional level remains elusive. 

Microribonucleic acids (miRNAs) are small non-coding RNA regulators of approximately 21 nucleotides that act at a post-transcriptional step of gene expression affecting diverse fundamental biological processes including cellular immune functions and development [18,19,20]. These non-coding RNAs inhibit protein synthesis by binding to the 3’-untranslated region (3’-UTR) of target mRNAs in a partially complementary fashion [21,22]. In silico prediction analysis shows that many miRNAs can interact with KIR3DL3, therefore the aim of the current study was to identify the miRNAs that downregulate KIR3DL3. 

Here, we provide evidence that the miRNAs miR-26a-5p, miR-26b-5p, and miR-185-5p control KIR3DL3 expression. This pathway may be genetically manipulated to increase broad cytotoxicity of NK cells through potential target miRNAs.

## 2. Materials and Methods 

### 2.1. Cell Culture

Three NK cell lines NK92, YT, and YT-KIR3DL3 and the K562 cell line were used in this study. YT-KIR3DL3 cells were developed by stable transfection of YT cells with a plasmid containing full-length KIR3DL3. These cells can constitutively express the KIR3DL3 protein on the surface [16]. YT-KIR3DL3 cells were kindly provided by Professor Ashley Moffet (University of Cambridge, Cambridge, UK). Cells were maintained in the medium Roswell Park Memorial Institute (RPMI) 1640 supplemented with 10% heat-inactivated fetal bovine serum, 2 mM L-glutamine, 100 U/mL penicillin, 100 μg/mL streptomycin (Gibco, Grand Island, NY, USA), 0.1 mM 2-mercaptoethanol and 100 U/mL recombinant human interleukin-2 (IL-2; Sigma, St. Louis, MO, USA) for IL-2-dependent NK92 cells. All cells were cultured at 37 °C with 5% CO_2_. 

### 2.2. Bioinformatic Prediction of Potential miRNAs for KIR3DL3

All 2,588 reported human miRNA data were obtained from miRBase (http://www.mirbase.org/) [23], and all allelic 3’-UTR sequences of KIR3DL3 were retrieved from the IPD-KIR Database (https://www.ebi.ac.uk/ipd/kir). To predict the potential miRNA that may regulate KIR3DL3 by acting on the 3’UTR of the gene, we utilized multiple prediction programs with inherently different algorithms, including RNAhybrid [24], Targetscan [25] and miRanda [26]. The candidate miRNAs were selected from three integrative searches.

### 2.3. Quantitative Real-Time Reverse Transcription Polymerase Chain Reaction (qRT-PCR)

Total RNAs, including miRNAs and mRNAs, were extracted using the Illustra RNAspin Mini RNA Isolation Kit (GE Healthcare Life Sciences, Buckinghamshire, Buckingham, UK), according to the manufacturer’s protocol. The RNAs were converted to cDNAs using the Moloney murine leukemia virus reverse transcriptase (M-MLV RT) (Promega, Madison, MD, USA) and random hexamers. In addition, reverse-transcription of mature miRNAs was performed using stem-loop reverse transcript primers [27,28]. 

Amplification of cDNA was performed using specific primers [29] (Appendix A) and the QuantiFast SYBR Green PCR kit (QIAGEN, Germantown, MD, USA) on an ABI PRISM 7500 Real-Time PCR system (Applied Biosystems, Foster City, MA, USA). RNA polymerase II (RPII) was used as an endogenous control for KIR3DL3 and miR-16 for miRNA. Relative gene expression was calculated as 2^ΔCT^: ΔC_T_ = C_T_ KIR3DL3 - C_T_ RPII. For miRNA detection, amplification was performed using specific forward primers and universal reverse primer [28] and relative expression calculated as 2^ΔCT^: ΔC_T_ = C_T_ miRNA - C_T_ miR-16.

### 2.4. Reporter Plasmid Construction and Luciferase Reporter Assay

Luciferase vectors, *firefly* and *renilla* plasmids (pcDNA3.1-Zeo(+)Pp and pRL-SV40, respectively), were kindly provided by Dr Yong Sun Lee, University of Texas Medical Branch, USA. Specific primers were used for the amplification of wild-type 3’UTR KIR3DL3 (p3UTR_WT) and site-specific miRNA mutagenesis (pMu_miR) (primer sequences in Appendix A). All fragments were amplified from the genomic DNA of NK92 and subsequently inserted between *Bam*HI and *Xho*I downstream sites of the *firefly* gene in pcDNA3.1-Zeo(+)Pp, leading to the modified constructs as shown in Table 1. The sequences of the constructs were confirmed by Sanger-based DNA sequencing (Macrogen Inc., Seoul, Korea).

In 24-well plates, 2.5 × 10^5^ YT cells were transiently co-transfected with 1.8 µg of the pcDNA3.1-Zeo(+)Pp reporter plasmid or modified pcDNA3.1-Zeo(+)Pp containing the 3’UTR of KIR3DL3, and 0.2 µg of the pRL-SV40 control vector encoding *renilla* luciferase (Promega, Madison, MD, USA) by using the FuGene^®^ HD transfection reagent (Roche, Indianapolis, MA, USA). After 48 h of transfection, luciferase activities of each construct were measured by the Dual-luciferase reporter assay (Promega, Madison, MD, USA) by the GloMax^®^ 20/20 luminometer machine (Promega, Madison, MD, USA). The *firefly* luciferase activity was normalized with the *renilla* luciferase activity and then normalized to the average activity of the control reporter. Experiments were performed in triplicates. The Paired Student’s *t*-test was used to compare the average luciferase activities in various reporter vectors [30].

### 2.5. Targeted miRNA Inhibition

The mirVana® miRNA inhibitors for miR-26a-5p, miR-26b-5p, and miR-185-5p (Ambion, Austin, TX, USA) were used in this study with a negative control or scramble control. K562 cells were transfected with 500 nM of inhibitors using the Neon® Transfection System (Invitrogen, Thermo Fisher Scientific Inc., Grand Island, NY, USA). The cells were then collected at 48 h after transfection and analyzed for the expression of mRNA and cell surface KIR3DL3 expression by flow cytometry. 

### 2.6. Flow Cytometry

Cells were stained with mouse IgG_2B_ isotype control (R&D Systems, Minneapolis, MI, USA) or primary mouse anti-KIR3DL3 antibody kindly provided by Professor Ashley Moffet, University of Cambridge, UK [16]. Where necessary, a goat anti-mouse PE secondary antibody was used (R&D Systems, Minneapolis, MI, USA). The level of KIR3DL3 expression was analyzed by the Beckman Coulter Gallios and the Kaluza Flow Analysis Software (Beckman Coulter, Fullerton, CA, USA). Percentage of positive cells and mean fluorescence intensity (MFI) were calculated following consideration of the values from the isotype control. 

### 2.7. Statistical Analysis

All data were presented as mean ± standard error of the mean of three independent experiments. All data were tested for normal distribution by the Shapiro–Wilk Test. The statistical differences between two groups were analyzed using the Student’s *t*-test by GraphPadPro Prism5.0 (GraphPad, San Diego, CA, USA). A value of *p* < 0.05 was considered a statistically significant difference, and the degree indicated as follows: *, *p* < 0.05; **, *p* < 0.01; ***, *p* < 0.001.

## 3. Results

### 3.1. Bioinformatic Prediction of Potential miRNA Candidates for KIR3DL3

Initially, KIR3DL3 expression was evaluated in the NK92 cell line to confirm regulation at the post-transcriptional level. NK92 cells had detectable mRNA for KIR2DL4, 3DL1, and 3DL3 but lacked cell surface expression of KIR3DL3 (Appendix A), supporting a post-transcription regulation mechanism.

Next, to identify the potential miRNAs targeting the 3’UTR of KIR3DL3, we used three bioinformatic programs RNAhybrid [24], TargetScan [25] and miRanda [26] to produce a target prediction list. Of the list of putative miRNA targets, 26 miRNAs were identified by all three programs (Figure 1a). In agreement with previous reports [31], six miRNAs were matched with the list of miRNAs expressed in human (CD56^+^CD3^-^) NK cells, namely miR-26a-5p, -26b-5p, -185-5p, -203a-3p, -372 and miR-373. To determine whether these miRNAs could contribute to the regulation of KIR3DL3 at the post-transcriptional step, NK92 and YT cells were analyzed for the expression of the six candidate miRNAs by Stem-loop qRT-PCR. Specifically, miR-26a-5p, -26b-5p and miR-185-5p were expressed in both NK cell lines, however miR-203a-3p was only expressed in NK92 cells, albeit at a low level (Figure 1b and Appendix A). Therefore, the three miRNAs expressed in all cell lines were selected as potential miRNA candidates regulating KIR3DL3.

### 3.2. Potential miRNA Candidates Targeting KIR3DL3 Through the 3’UTR 

To investigate the role of specific miRNAs binding to the 3’UTR of KIR3DL3, we constructed a panel of reporter plasmids containing wild-type and mutated miRNA binding sites of the KIR3DL3 3’UTR (Table 1). Site-directed mutagenesis was used to produce one plasmid containing a mutated site for each miRNA, and a second encompassing combinations of mutations (>2 sites) to investigate synergistic effects of the miRNAs. For the miRNAs miR-26a-5p and 26b-5p they were identical at the seed sequences and had two target sites on the 3’UTR. The miRNA miR-203a-3p was present at a very low level in NK92 cells or absent in YT cells (as described above) and accordingly the mutation plasmid for the miR-203a-3p binding site (pMu_miR-203a-3p) acted as a control such that in the case of low level or absent expression of the miRNA, mutating the miRNA binding site should have no effect. These plasmid constructs were transfected into YT cells. As shown in Figure 2, luciferase activity was significantly decreased to 60% for the plasmid with the wild type 3’UTR (p3UTR_WT) compared to the control (empty) vector. Moreover, point mutations at both binding sites for miR-26a/b-5p (pMu_miR-26a/b-5p (295), pMu_miR-26a/b-5p (344)) and for miR-185-5p (pMu_ miR-185-5p), induced recovery of the reporter gene expression, while two site mutations for miR-26a/b-5p (pMu_miR-26a/b-5p (295,344)) also up-regulated luciferase levels, albeit without the synergistic effect. As expected, there was no difference in the luciferase activity in the mutated miR-203a-3p binding sites (pMu_miR-203a-3p) resulting from the absence of expression of this miRNA in YT cells.

### 3.3. Knocking Down Candidate miRNAs Resulted in Higher Cell Surface KIR3DL3 Expression

We determined the influence of candidate miRNAs on the cell surface expression of KIR3DL3. K562, an erythroleukemia cell line with low surface expression of KIR3DL3, was transfected with miRNA inhibitors for miR-26a-5p, miR-26b-5p and miR-185-5p and in combination (miR-26a/b-5p or miR-26a/b-5p ± 185-5p) leading to elevated KIR3DL3 cell surface expression (Figure 3). Compared with the scramble control, endogenous miR-26a-5p, miR-26b-5p, and miR-185-5p were significantly decreased in the miRNA inhibitor transfected K562 cells (Figure 3a). As expected, specifically knocking down miRNAs caused upregulation of the cell surface KIR3DL3 on K562 cells. When compared to negative control (scramble control), inhibiting effect of miR-26a-5p, miR-26b-5p and miR-185-5p significantly increased surface KIR3DL3 expression for both mean fluorescent intensity and percentage of positive cells (Figure 3c,d). In addition, the synergistic effects of miR-26a-5p and miR-26b-5p and three miRNA inhibitors (miR-26a/b-5p ± 185-5p) also displayed a strong regulation effect but was not significantly different compared to a single miRNA inhibition (miR-26b-5p and miR-185-5p). However, the regulation efficiency of miR-26a-5p was not as great as miR-26b-5p and miR-185-5p. Consistent with increased protein expression, significantly decreased endogenous mRNA levels were observed in cells after exposure to antisense miRNAs, except for anti-miR-26a-5p (Figure 3b).

## 4. Discussion

It has been unclear how KIR repertoire is created for NK cell clonality. Expression of KIR3DL3 has been proposed to be regulated by DNA methylation [16]. A recent study showed that KIR3DL3 mRNA was present in purified decidual NK and also peripheral blood NK cells but the corresponding protein was not detected at the cell surface [14,16]. Therefore, expression of KIR3DL3 may be inducible in developmental situations or in some diseases. In this study, our data revealed for the first time that KIR3DL3 is also regulated by miRNAs. These results provide a molecular rationale for undetectable KIR3DL3 expression on the cell surface. 

Analyses of gene regulation have shown that miRNAs play an important role in controlling many critical cellular processes in NK cells. For example, miR-181 and miR-150 have been found to influence NK cell function, maturation, development, and cell survival [31,32,33]. Furthermore, the presence of miR-1245 regulates the expression of the NK cell activating receptor, NKG2D [34]. In this study, we used three different target prediction software programs (RNAhybrid, TargetScan, and miRanda) to identify 26 candidate miRNAs that could bind the 3’UTR of KIR3DL3. To refine our analysis, we used previous data from microarray studies that identified a number of miRNAs in NK cells [31] and our own screen of miRNAs in NK cell lines, NK92 and YT, to restrict our validation to three potential miRNAs (miR-26a-5p, -26b-5p and -185-5p) that could modulate surface KIR3DL3 expression. The miRNA miR-34a-5p in the bioinformatic prediction was not included in our analysis as although it has been shown to be up-regulated in immature CD56^bright^ NK populations and has been proposed to regulate KIR3DL3 expression [18]. 

Normally, the KIR genotype should be the main determinant of receptor expression on NK cells. Thus, KIR repertoire and NK clonality should be generated by gene regulation. The gene sequence of KIR3DL3 is very similar to those of other KIRs, albeit it is regulated independently [12,16]. Most studies proposed an integrated model for the stochastic expression of KIR by DNA methylation [10,35]. In that, KIR3DL3 displays methylation at the 5’ promotor region [16] and exposure to a methyltransferase inhibitor effectively induced KIR3DL3 expression on NK92 cells [16]. In addition to DNA methylation, here, we showed that the knocking down of candidate miRNAs effectively induced KIR3DL3 protein cell surface expression. 

High sequence identity is a common feature of the KIR gene family [36] and the 3’UTR is no exception. We analyzed the sequences of the 3’UTR of the KIR genes and found that KIR3DL3 has a high degree of sequence identity with the 3’UTR of other KIRs (86–95%) as shown in Appendix A. All three miRNA candidates could bind to 3UTR of all 14 KIR genes (Appendix A). These data suggest that modulation of KIRs could be dependent on miRNAs. In addition, it is reasonable to believe that KIR repertoire in different individual clonal NK cells reflects the presence of differential miRNA expression. The diverse KIR receptor patterns consistently correlate with the different stages of NK maturation and their immune responses. 

Theoretically, anti-sense of miRNA should not affect mRNA but only miRNA expression. However, in our experiments, we found that the transfection of anti-miRNAs could also reduce the level of KIR3DL3 mRNA (Figure 3b) except anti-miR26a-5p. This is probably due to the kinetic of mRNA and protein turn-over, reducing the mRNA level. This phenomenon has also been observed by others [37]. Interestingly, the observation for anti-miR26a-5p reflects only a direct effect on the expression of this miRNA with a corresponding increase in cell surface expression, albeit, it was the lowest level compared to the others.

K562 cells expressing a low level of endogenous mRNA and protein of KIR3DL3 was used for functional verification and validation by knocking-down the candidate miRNAs. The surface KIR3DL3 was significantly upregulated represented by the percentage of positive cells and mean fluorescent intensity (MFI). Interestingly, there were synergistic effects observed from multiple miRNAs, especially the three-miR combination. Moreover, different miRNAs differed in the degree they modulated expression of KIR3DL3. Unfortunately, the analysis of gain of function approach by transfection with the mimic candidate miRNAs was not possible because the over-expression of miRNA candidates led to cell death (data not shown). It was recently observed that over-expression of miR-185-5p induced apoptosis by directly targeting on anti-apoptotic genes, B-cell lymphoma (BCL)2 and BCL2L1, in prostate cancer cell lines [38] as well as the effect of miR-26a-5p [39]. In addition, miR-26a/b is a potential autophagy inhibitor in hepatocellular carcinoma (HCC) to inhibit the expression of serine/threonine protein kinase ULK1 promoting cell apoptosis [40]. Therefore, overexpression of miR-26a-5p, -26b-5p and -185-5p might cause toxicity to cells. These observations were suggestive by most putative gene targets exclusively intermediated in apoptotic pathways by these candidate miRNAs (Appendix A). 

A limitation of this study was that in our hands the NK92 cell line was resistant to transfection and, therefore, the YT cell line was used to evaluate the miRNA-mRNA interactions by the dual-luciferase reporter assay. However, initial data analysis suggested that the 3’UTR of KIR3DL3 was a potential target for three miRNAs (miR-26a-5p, -26b-5p and -185-5p) expressed in YT cells (Appendix A).

In summary, we have identified for the first time a miRNA regulatory mechanism that modulates KIR3DL3 surface expression. This mechanism could contribute to and maintain clonal expression patterns of KIRs on immune cells relevant to immune responses. The information could assist in further studying the function and ligand identification of KIR3DL3.

## Figures and Tables

**Figure 1 genes-10-00603-f001:**
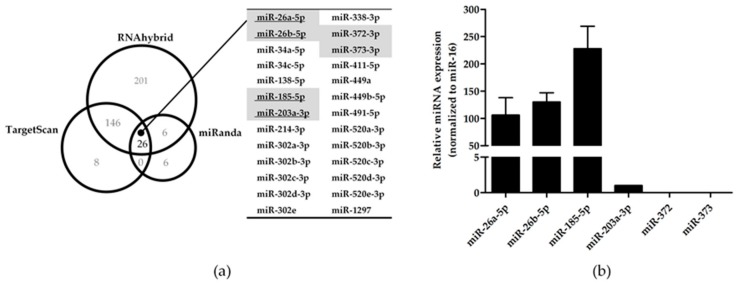
A summary of candidate miRNA prediction targets in KIR3DL3 and the endogenous expression of specific miRNAs. (**a**) Venn diagram representing the number of potential miRNAs for KIR3DL3 and the common miRNAs (*n* = 26) predicted from three different miRNA-target prediction software (RNAhybrid, TargetScan and miRanda). The table highlights six miRNAs matched with previously identified miRNAs in human (CD56^+^CD3^-^) NK cells [31]. Only four miRNA candidates were examined in the NK-92 cell line (underlined). (**b**) Endogenous expression of six miRNAs in NK-92 cells was determined by quantitative real-time polymerase chain reaction (qRT-PCR). The expression levels were normalized to miR-16. The results are depicted as mean ± standard error of mean (S.E.M).

**Figure 2 genes-10-00603-f002:**
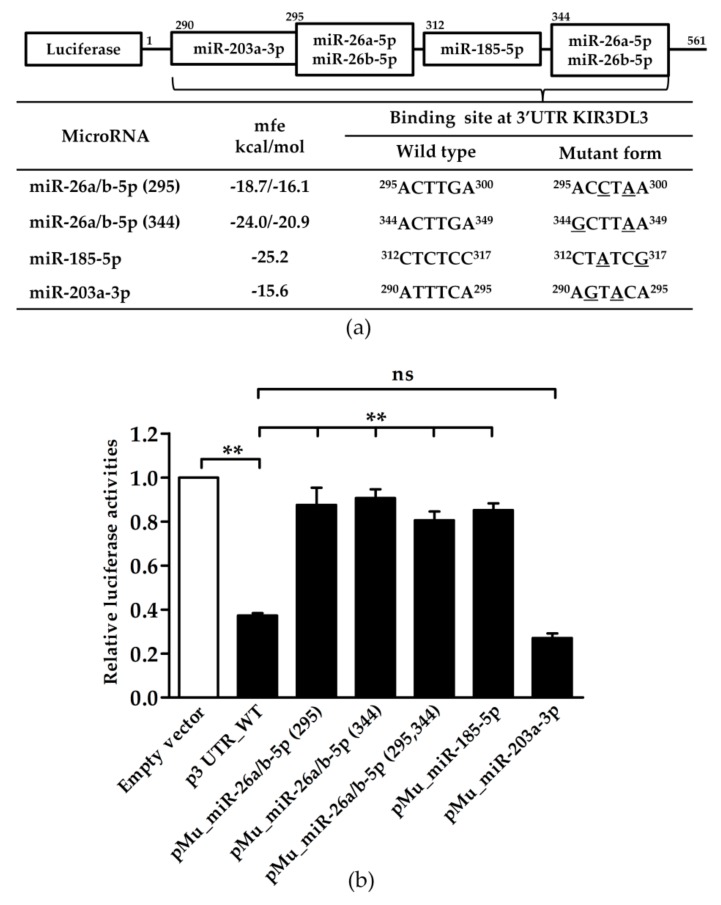
The effect of endogenous miRNAs interacting with the 3’UTR of KIR3DL3 as indicated by luciferase activity. (**a**) The predicted miRNA binding sites on the 3’UTR of KIR3DL3 are presented in the table. The minimum free energy (mfe) for the binding of each miRNA was calculated by the RNA hybrid program. MiRNA binding sites were mutated by PCR directed mutagenesis and were confirmed by DNA sequencing as shown in the table. The superscript numbers represent the position of the seed sequence binding to the 3’UTR of KIR3DL3. (**b**) Results from the dual-luciferase assay in YT cell lines when transfected with luciferase constructs of the 3’UTR wild-type (p3UTR_WT) or mutant forms (mutated miRNA-binding site, pMu_miR). The *firefly* luciferase activity was normalized by *renilla* luciferase activity and then relative to empty vector. Data are representative of three independent experiments (mean ± S.E.M.). ** *p* < 0.01 (Student’s *t*-test).

**Figure 3 genes-10-00603-f003:**
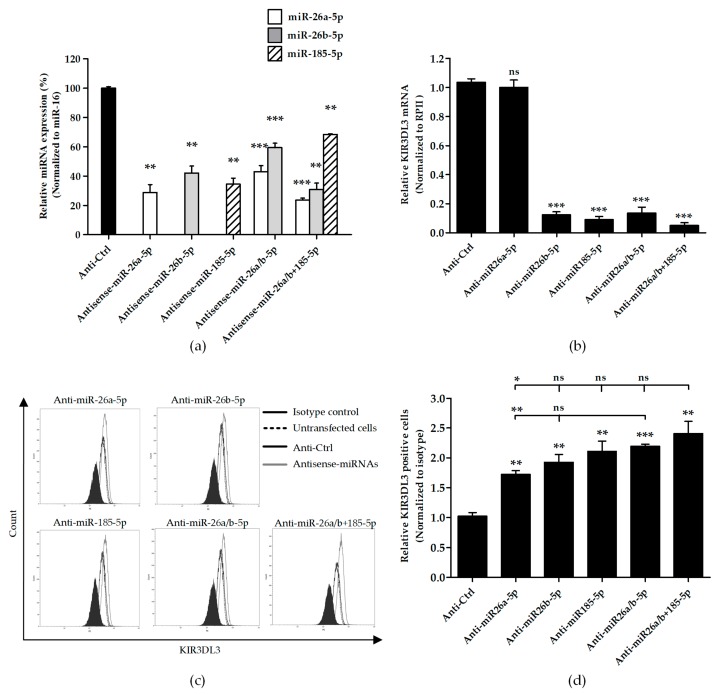
Knockdown candidate miRNAs increase the expression of KIR3DL3. K562 cells were transfected with 500 nM candidate antisense miRNAs for 48 h and subsequently assessed for KIR3DL3 expression. (**a**) Relative expression levels of mature miR-26a-5p, -26b-5p and 185-5p normalized to miR-16 when were compared to the anti-sense control (scramble) are presented. (**b**) The relative KIR3DL3 mRNA expressions are presented. Total RNA was extracted, and the levels of KIR3DL3 mRNA normalized to RPII were measured by RT-PCR. The figure shows the means ± S.E.M. from three independent experiments. (**c**) After transient transfection, cells were examined by flow cytometry for cell surface KIR3DL3 expression. The filled histogram represents cells stained with the isotype antibody. In the open histogram, the dotted line indicates K562 untransfected cells, the solid line shows the data from the negative control (anti-Ctrl or miRNA scramble control) and the gray line represents the transfection of 500 nM antisense miRNAs all stained with the KIR3DL3 antibody. (**d**) A relative number of positive cells for KIR3DL3 are shown. Results are calculated from three experiments and depicted as mean ± S.E.M. *, *p* < 0.05; **, *p* < 0.01; ***, *p* < 0.001.

**Table 1 genes-10-00603-t001:** List of plasmid constructs.

Plasmid Names	Information
p3’UTR_WT	Wild type 3’UTR of KIR3DL3
pMu_miR-26a/b-5p (295)	Mutated miR-26a/b-5p binding site 3’UTR at positions 297 and 299
pMu_miR-26a/b-5p (344)	Mutated miR-26a/b-5p binding site 3’UTR at positions 344 and 348
pMu_miR-26a/b-5p (295,344)	Mutated both miR-26a-5p and miR-26b-5p binding sites
pMu_miR-185-5p	Mutated miR-185-5p binding site
pMu_miR-203a-3p	Mutated miR-203a-3p binding site

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
