# Peer review of "Regulation of KIR3DL3 Expression via miRNA"

_genes, 2019, doi:10.3390/genes10080603_

Round 1

Reviewer 1 Report

A nice piece of work on the very interesting topic of KIR receptors. I have a question on Supp S3. From which cell data were these data on other KIRs extracted.? 

Furthermore I would suggest the authors to gather and show data on the binding sites of miR-26a/b and miR-185 in the other KIRs 3'UTR. Those that also have binding sites for these miRNAs should be tested for expression and compared with those with no binding sites 

Author Response

Reviewer 1

Point 1: I have a question on Supp S3. From which cell data were these data on other KIRs extracted?

Response 1: All allelic 3’-UTR sequences of KIRs were retrieved from a repository of the IPD-KIR Database (https://www.ebi.ac.uk/ipd/kir) which reported sequences published from NK cells and stored in the database. To avoid the confusion, we have added this information to the footnote of the supplementary table S3.

Point 2: Furthermore I would suggest the authors to gather and show data on the binding sites of miR-26a/b and miR-185 in the other KIRs 3'UTR. Those that also have binding sites for these miRNAs should be tested for expression and compared with those with no binding sites.

Response 2: The three candidate miRNAs (miR-26a/b and miR-185) can bind to the 3’UTR of all 14 KIR genes as shown in Supplementary Table S4 below because the sequences of 3’UTR of KIRs have high sequence identities as mentioned in Supplementary Table S3. We agree with the reviewer that bioinformatic analysis need to be functionally tested. However, testing with other KIR genes is beyond the scope of this manuscript. Thus, we would prefer to present the functional data related to this issue in a different manuscript. To clarify the reviewer’s concern, we have added to lines 266-267 to read “All 3 miRNA candidates (miR-26a/b and miR-185) could bind to 3UTR of all 14 KIR genes (Supplementary S4). Accordingly, the Supplementary S4 has now been changed to S5.

Supplementary Table S4 Computational prediction of three candidate miRNAs binding on 3’UTR of 14 KIR genes.

KIRs

mfe (kcal/mol)a

miR-26a-5p

miR-26b-5p

miR-185-5p

2DL1*0020101

18.7/-24.0

-16.1/-20.9

-25.2

2DL2*0010101

-19.0/-24.0

-16.4/-20.9

-25.2

2DL3*0010101

-18.7/-24.0

-16.1/-20.9

-19.8/-25.2

2DL4*0010201

-18.7/-24.0

-16.1/-20.9

-22.5/-25.2

2DL5A*0010101

-18.7/-24.0

-16.1/-20.9

-25.2

2DS1*0020101

-18.7

-16.1

-25.2

2DS2*0010101

-18.7

-16.1

-25.2

2DS3*0010301

-18.7/-24.0

-16.1/-20.9

-25.2

2DS4*0010101

-18.7

-16.1

-25.2

2DS5*0020101

-18.7/-24.0

-16.1/-20.9

-25.2

3DL1*0010101

-19.0/-24.0

-16.4/-20.9

-25.2

3DL2*0010101

-17.9/-23.9

-15.0/-21.0

-25.2

3DL3*00101

-18.7//24.0

-16.1/-20.9

21.4/-25.2

3DS1*0130101

-19.0/-24.0

-16.4/-20.9

-25.2

a minimum free energy (mfe) of miRNA:mRNA duplex was calculated using RNAhybrid

Reviewer 2 Report

The author started from the observation that KIR3DL3 was transcribed but not translated in NK cells.They predicated and confirmed that three miRNAs regulated the expression of KIR3DL3. The overall study is interesting, but there are several concerns that need to be addressed.

1) miRNAs regulates gene expression at transcriptional and/or post-transcriptional level. The major concern here was the author stated that the three miRNAs regulated KIR3DL3 post-transcriptionally. But as they show here in Figure 3b, the antisense oligos decreased the mRNA levels, except for miRNA-26a. That means at least for miRNA-26b and miRNA-185, the regulation was at transcriptional levels. Please clarify this.

2) The antisense oligos may have off-target. The author should use CRISPR know-down or knock-out to confirm their conclusion.

3) The significance of the study was limited with just expression regulation study, while no results on miRNAs’ function on NK cells? Will NK cell’s function be affected by the three miRNAs?

4) As the author introduced in the introduction, it has been proposed that KIR3DL3 is expressed in specific pathological situations or during development. The author can evaluate the expression of the three miRNAs and KIR3DL3 at different situations to investigate if there is a correlation.

5)  Line 152, there is one typo, miRNA-37 should be miRNA-373.

Author Response

Reviewer 2

Point 1: miRNAs regulates gene expression at transcriptional and/or post-transcriptional level. The major concern here was the author stated that the three miRNAs regulated KIR3DL3 post-transcriptionally. But as they show here in Figure 3b, the antisense oligos decreased the mRNA levels, except for miRNA-26a. That means at least for miRNA-26b and miRNA-185, the regulation was at transcriptional levels. Please clarify this.

Response 1: We agree with the reviewer’s notion. Anti-sense to miRNA should not result in reduction of target mRNA. This observation has also been reported by others as explained in lines 272-278. We envisage that when the miRNA is bound to anti-sense, freeing mRNA for protein translation. If the mRNA production is not efficient, this should lead to the reduction of mRNA. We use the term “the kinetic of mRNA and protein turn-over” which is on lines 274-275.

Point 2: The antisense oligos may have off-target. The author should use CRISPR know-down or knock-out to confirm their conclusion.

Response 2: We agree with the reviewer that CRISPR is a better technology than anti-sense. However, most of miRNA studies have used anti-sense for down regulation of miRNA. Apparently, although it is not efficient, the effect of anti-sense was clearly demonstrated, the conclusion could be drawn and presented in this manuscript.

Point 3: The significance of the study was limited with just expression regulation study, while no results on miRNAs’ function on NK cells? Will NK cell’s function be affected by the three miRNAs?

Response 3: The function of KIR3DL3 is not well defined. In addition, its ligand has not been identified. It would be difficult to design an experiment on function. By knowing the mechanism of how to up-regulate the gene, would lead to further investigations i.e. ligand identification and function.

Point 4: As the author introduced in the introduction, it has been proposed that KIR3DL3 is expressed in specific pathological situations or during development. The author can evaluate the expression of the three miRNAs and KIR3DL3 at different situations to investigate if there is a correlation.

Response 4: Thank you for the suggestions. We are in the process of further investigations

Point 5: Line 152, there is one typo, miRNA-37 should be miRNA-373.

Response 5: We have edited it. Thank you so much. It is now on line 155.

To the Editor

Since the submission, there has been an updated on miRNA nomenclature. Thus, we have updated the names of miRNAs in Figure 1. The Figure 1 has been replaced with the new one.

Round 2

Reviewer 2 Report

The authors have appropriately addressed all the comments.